# Wave storm dynamics and clustering, and their impacts on beach erosion

Salika Thilakarathne[1] , Hesamodin Enayatighadikolaei[2] , Md Shofiqul Islam[3] , Takayuki Suzuki[2] , Martin Mäll[2] and Ralf Weisse[1]

[1]Institute of Coastal Systems - Analysis and Modeling, Helmholtz-Zentrum Hereon, Geesthacht, Germany; [2]Department of Civil Engineering, Yokohama National University, Yokohama, Japan and [3]Institute of Disaster Management, Khulna University of Engineering & Technology, Khulna, Bangladesh

## Research Article

**Keywords:**
coastal change; wave storms; beach erosion; machine learning; seasonal predictions

**Corresponding author:**
Md Shofiqul Islam;
Email: msislam@idm.kuet.ac.bd

S.T., H.E., M.S.I. are joint first authors.

## Abstract

We analyse a 36-year hydrodynamic and morphological dataset from the Hasaki coast, Japan, comprising 501 wave storm events (405 individual and 96 clustered events) to investigate the impact of storm dynamics and clustering on beach erosion. Focusing on the wave component of storms, events are identified using wave height thresholds. Daily and weekly beach profile measurements from the Hasaki Oceanographic Research Station are used to quantify erosion. The study examines the seasonal influences on Hasaki beach, the characteristics and temporal evolution of storms, and their associated erosional impacts. Moreover, we test two supervised machine learning (ML) algorithms, support vector regression (SVR), and deep neural network (DNN), in predicting shoreline change using 16 wave, storm, and morphological features. SVR showed reasonable accuracy on the training dataset but underperformed on testing, while DNN failed to produce reliable predictions on both. With SVR yielding an $R^2$ of 0.18 and DNN 0.27 on the testing dataset, we conclude that, given the limited data and available features, such ML models may not generalise well. However, separate analyses using observed data reveal clear seasonal variations in wave storm dynamics and distinct behaviours of clustered events associated with beach erosion, highlighting important insights beyond the ML results.

## Impact statement

Development of tools and methods for projecting coastal erosion is increasingly important for sustainable coastal management and planning practices. The need for such tools is largely accelerated by the climate change uncertainty and its possible impacts on any given coastal zone through wave storm activity (i.e. local threshold exceeding wave heights). Wave storms tend to have a seasonal activity that drives the majority of erosional events, and these can be further classified into single and clustered events. Beach response to these storm waves depends not only on storm strength but also on the temporal timing (length and time of year), as well as other conditions. This study aimed to develop first insights into the development of such tools through machine learning (ML) techniques by using a 36-year hydrodynamic and morphological dataset from the Hasaki coast, Japan. The findings show that the current ML framework applied underperforms in estimating shoreline response and requires further work for both the techniques as well as more comprehensive data considerations. This also suggests that more open, standardised, and accessible coastal datasets are necessary to build reliable and generalisable ML tools. The insights from this work can guide the further development of impact-based coastal monitoring and predictive systems that reflect seasonal and storm clustering effects on beach evolution.

## Introduction

Climate change is amplifying the frequency and intensity of extreme weather events, increasing the vulnerability of coastal systems and calling for urgent, informed mitigation strategies. Among storm-induced disturbances, beach erosion poses a major threat to both natural coastal environments and built infrastructure. Although beaches are continuously shaped by waves, tides, and winds, storms remain the dominant drivers of abrupt and significant morphological change (Birkemeier, 1979). Approximately 24% of sandy beaches worldwide are already in erosionary states (Luijendijk et al., 2018), and this trend is expected to worsen under future climate scenarios (Vousdoukas et al., 2020). While the impacts of individual storms are well documented, storm clustering (successive storms occurring within short timeframes) remains underexamined, despite their potential for causing more sustained erosion due to cumulative energy input and limited recovery intervals.

The increasing intensity and frequency of coastal storms are particularly concerning given the growing population density and ongoing economic development in coastal areas (Small and Nicholls, 2003; Neumann et al., 2015; Kulp and Strauss, 2019). These events, often driven by atmospheric extremes such as extratropical and tropical cyclones, intensify coastal hazards through wave impact and overwash (Rivas et al., 2022). While storms comprise multiple interacting components, such as storm surge, wind, rainfall, and waves, the present study focuses specifically on the wave component, which is generally the dominant driver of storm-induced erosion along open sandy coasts (Hansen and Barnard, 2010; Splinter et al., 2014; Ahmad et al., 2015; Lobeto et al., 2024). Recent studies suggest that clusters of moderate wave storms can inflict more damage than a single storm with similar characteristics (Coco et al., 2014; Karunarathna et al., 2014; Dissanayake et al., 2015; Masselink et al., 2016). This enhanced erosive effect is often linked to progressive sediment loss and inadequate time for beach recovery between events (Eichentopf et al., 2020). However, the physical processes underlying the impacts of clustered events remain poorly quantified, highlighting the need for detailed morphological and hydrodynamic analyses.

Effective coastal management, especially in wave storm-prone and seasonally dynamic regions, requires a clear understanding of how beaches respond to both individual and clustered events (Masselink et al., 2016). While short-term storm-induced morphological changes can be substantial, medium- to long-term coastal evolution is often governed by seasonal and interannual variability (Suzuki and Kuriyama, 2007, 2014; Hansen and Barnard, 2010; Kuriyama et al., 2012; Pianca et al., 2015). For instance, the shoreline dynamics of Japan's main island, Honshu – bordered by the Sea of Japan, the Philippine Sea, and the North Pacific Ocean – are influenced by a range of extreme wave conditions, including typhoons, extratropical cyclones, and locally generated storm events (Dorman et al., 2004; Eichentopf et al., 2020; Shimozono et al., 2020; Suzuki et al., 2020). Long-term observations have revealed patterns of shoreline oscillation and storm-driven variability that are strongly modulated by these seasonal conditions (Eichentopf et al., 2020). Similarly, seasonal changes in beach profiles have been linked to the frequency and intensity of storm activity in other coastal regions (Angnuureng et al., 2017). These findings highlight the importance of integrating high-resolution storm data with long-term beach morphology observations for robust risk assessments and accurate seasonal-scale impact prediction (Bird, 2008; Vousdoukas et al., 2012).

Recent rapid developments in machine learning (ML) algorithms, combined with the increasing availability of large datasets, have led to a significant rise in ML-aided studies in coastal sciences (Goldstein et al., 2019; Beuzen and Splinter, 2020; Ellenson et al., 2020; Hayuningsih et al., 2024; Tabasi et al., 2025). Due to their excellent ability to fit models to available datasets while maintaining strong generalisation capability – although this often depends on the size and quality of the training data – ML-aided approaches are increasingly applied to understand and predict coastal extreme events. Wave storm-induced beach erosion prediction studies, in particular, have benefited from these data-driven approaches. For example, Thilakarathne et al. (2024a,b) demonstrated how ML and explainable artificial intelligence (AI) techniques can reveal the relative contributions of morphological and hydrodynamic features to beach erosion predictions. Such insights offer promising new avenues for advancing coastal process understanding. However, challenges remain, such as the limited availability of high-quality datasets that often hampers the training of models with robust generalisation ability, as highlighted by Miller et al. (2019). Despite these limitations, ML-based approaches are rapidly expanding, especially for coastal extreme impacts, underscoring both their potential and the need for further methodological improvements.

As groundwork for an upcoming research project on seasonal-scale coastal impact prediction, this study investigates the evolution of wave storm clusters and evaluates whether their behaviour and impacts differ from those of individual events. We use a data-driven approach to assess how storm characteristics, hydrodynamic conditions, and initial shoreline position influence shoreline change ($dSL$). Statistical analyses and ML techniques are applied to identify key drivers of storm-induced morphological change and characterise the evolution of clustered storm behaviour. This study also seeks to advance understanding of the mechanisms and impacts of clustered storms and evaluate their significance for seasonal-scale shoreline change predictions. The structure of the paper is as follows: Materials and methods section outlines the materials and methods; Results and discussion section presents the results along with a discussion of their implications for predictive modelling, including limitations of the ML approach and recommendations for future impact-based seasonal prediction studies; and Conclusion section concludes with key findings.

## Materials and methods

### Study site

Hasaki coast, located in Kamisu City of Ibaraki Prefecture, Japan, features a nearly straight sandy beach approximately 16 km long, facing the North Pacific Ocean (Figure 1). The beach is characterised by medium-sized sediment with a diameter of approximately 0.18 mm (Katoh, 1995; Gunaratna et al., 2019), showing minimal variation along the cross-shore profile (Katoh, 1995). Due to the absence of significant alongshore bathymetric feature variations, it is classified as a longshore uniform beach (Kuriyama, 2002). With a tidal range of 1.45 m – where high, mean, and low water levels are 1.25, 0.65, and −0.20 m, respectively – and a mild

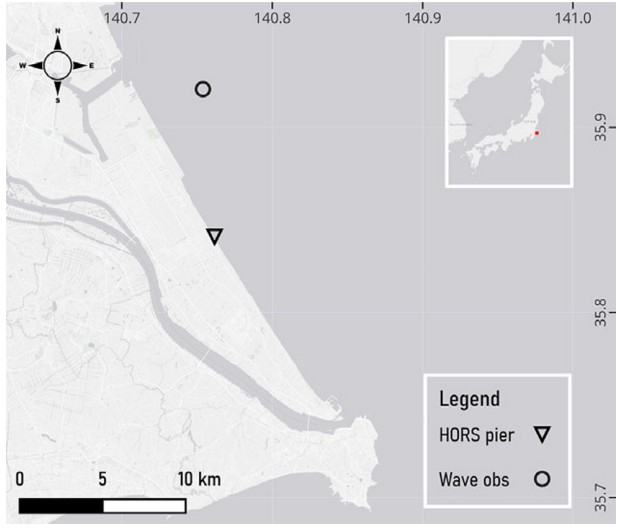

**Figure 1.** Location of the Hasaki coast in Kamisu City, Ibaraki Prefecture, Japan. The offshore wave gauge and the Hasaki Oceanographical Research Station (HORS) pier locations are marked on the map. Additional photographs and details about the research station are available on the Port and Airport Research Institute website (Port and Airport Research Institute, 2025).

beach slope, Hasaki is considered a microtidal dissipative beach (Kuriyama, 2002). Based on long-term wave characteristics, Banno et al. (2020) reported that wave conditions at Hasaki are generally more energetic in February, March, and October due to extratropical cyclones and late-summer typhoons. They also observed that wave periods are slightly longer in August due to typhoon-generated swells, while swell effects in September are less evident in monthly mean values despite frequent typhoon activity.

## Wave observations

We use two-hourly wave observations from the Nationwide Ocean Wave information network for Ports and Harbours (NOWPHAS) over the period 1987–2022. These data were obtained from a seabed-mounted ultrasonic wave gauge at a water depth of 24 m off Kashima Port (location shown in Figure 1). Wave direction is measured by a Doppler-type directional wave meter; however, wave directional data ($\theta$) are only available from July 1991 onwards. For missing directional values, we substitute the long-term mean wave direction. The annual mean significant wave height ($H_s$) and period ($T_s$) are 1.32 m and 9.07 s, respectively. In addition to significant wave data, we also use mean wave height and period ($H_0$ and $T_0$), the highest one-tenth wave height and period ($H_{1/10}$ and $T_{1/10}$), and the maximum wave height and period ($H_{peak}$ and $T_{peak}$), all of which are pre-processed to address outliers and missing data. Outliers are identified using the interquartile range method, supplemented by expert judgement specific to the study site, while missing data are excluded based on NOWPHAS guidelines.

## Storm dataset

Site-specific storm thresholds are defined using a combination of minimum wave heights, minimum storm duration, and meteorological independence criteria, following the methods proposed by Harley (2017) and Nagai and Ogawa (2004). Nagai and Ogawa (2004) present site-specific wave height thresholds of 1.5 and 2.5 m for Hasaki, with threshold values varying depending on shoreline characteristics across Japan. However, they do not specify criteria for minimum storm duration or inter-event time gaps. Therefore, we adopt the statistical approach of Harley (2017) and define a minimum duration of 6 h and a meteorological independence criterion of 48 h. This 48-h interval balances the frequent occurrence of storms at Hasaki, avoiding overly broad clustering that masks individual storm effects, while also ensuring sufficient separation between events to assess their impacts. A storm must therefore last at least 6 h with $H_s$ exceeding 1.5 m and include at least one instance where $H_s$ exceeds 2.5 m (Thilakarathne et al., 2023, 2024a). When the time gap between two or more storm events is less than 48 h, we consider them as one clustered event. Figure 2a shows examples of an individual storm event and a clustered storm event, the latter comprising three storm pulses (May–June 2018). Applying these criteria yields 405 individual storm events and 96 clustered storm events. The temporal distribution of these events, along with the number of storm pulses in clustered events, is shown in Figure 2b. Due to 99% of the $H_s$ data being missing between 2008 and 2013, only seven storm events are identified during this period. As our temporal evolution analysis relies on annual and 3-year mean values, these events are excluded from the analysis of storm evolution and associated impacts.

Following storm event identification, wave and storm characteristic datasets are compiled. A total of 16 features are used in the ML approach to predict $dSL$. These include 11 wave parameters,

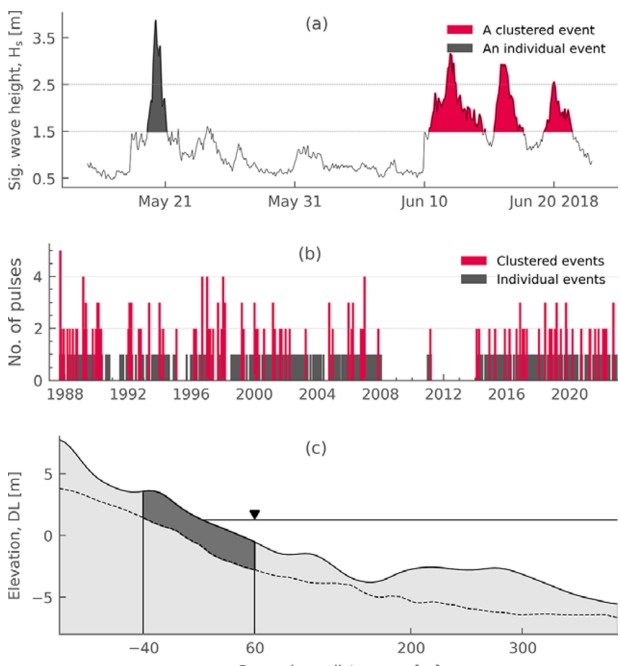

**Figure 2.** Key elements of the data processing workflow: (a) Storm identification using a two-threshold approach based on significant wave height ($H_s$). The selected period (May–June 2018) shows one individual and one clustered storm event (the latter comprising three pulses); (b) chronological distribution of identified individual and clustered storm events from 1987 to 2022, with the y-axis indicating the number of pulses per event (individual storms consist of a single pulse); and (c) example cross-shore profile from the HORS, showing the beach zone (−40 m to +60 m) used for volumetric erosion calculations. The dotted line represents the minimum recorded elevation and serves as the datum for estimating daily beach volumes within the shaded area, from which volumetric beach change ($dV$) is calculated. Elevations are referenced to the Hasaki datum level, and cross-shore coordinates follow the Hasaki coordinate system.

the month of the storm, storm duration, pre-storm shoreline position, and a storm power index. Moreover, the number of storm pulses is included as a predictor to represent storm clustering characteristics, where a single pulse indicates an individual event and multiple pulses represent the number of storms within a clustered event. The storm power index is calculated using methods adapted from Dolan and Davis (1994) and Karunarathna et al. (2014), as shown in Equation 1.

$$SP = H_{s-max}^2 \cdot D \tag{1}$$

where $SP$ is the wave storm power, $H_{s-max}$ is the maximum significant wave height, and $D$ is the storm duration (i.e. the time during which $H_s$ exceeds the 1.5 m threshold, for both individual and clustered wave storms).

## Cross-shore profile observations

We use 500 m long cross-shore profiles, recorded daily from January 1987 to March 2011 and weekly thereafter, to quantify shoreline and beach changes following storm events. These profiles are obtained by the HORS (Banno et al., 2020), which conducts surveys along a 427-m-long pier at 5-m intervals. The local coordinate system, ranging from −115 m (landward) to 385 m (seaward), is adopted for the present analysis. Profile elevations are referenced to the Hasaki datum level (Tokyo Peil −0.687 m) (Kuriyama, 2002).

The shoreline position at high water level (1.252 m) fluctuates between $-35.67$ and $45.76$ m. Accordingly, we define the beach section between $-40$ and $60$ m, based on Hasaki coordinates, for calculating volumetric beach changes ($dV$) (Figure 2c). First, we calculate the daily beach section volumes of the shaded area in Figure 2c, using the dotted line, which represents the minimum recorded elevation, as the datum. Shoreline change ($dSL$) and $dV$ are then calculated using pre- and post-storm shoreline positions and beach volumes. Erosional values are considered positive, while negative values indicate accretionary events.

### Machine learning models and training

We train both ML models using 16 predictors (input features), with the only predictand (target output) being $dSL$, which shows a strong correlation with $dV$ (see Supplementary Figure A1). While the dataset includes 501 storm events, substantial for coastal studies, it remains relatively small by machine learning standards. We use 80% of the data (400 events: 324 individual and 76 clustered) for training and the remaining 20% (101 events: 81 individual and 20 clustered) for testing, applying a random data-splitting function from the scikit-learn library (Pedregosa et al., 2011).

### Support vector regression model setup

The support vector regression (SVR) algorithm operates by mapping inputs into a high-dimensional space, using kernels such as radial basis function (rbf) and polynomial (poly) (Awad and Khanna, 2015). It constructs a hyperplane that minimises deviations within a specified error margin (epsilon). The model emphasises support vectors, which are critical training points near the margin, to ensure robust generalisation. The regularisation hyperparameter (C) controls error tolerance, while epsilon defines the permissible prediction error range. SVR is well-suited for relatively small datasets, making it an appropriate choice for the present study, which includes only 501 storm events. To optimise hyperparameters for balancing model complexity and prediction accuracy, we employ a grid search approach with 5-fold cross-validation. This method involves systematically evaluating 180 combinations of kernel types (rbf, poly), regularisation strengths (C: 0.1, 1, 5, 10, 25, 100), error margins (epsilon: 0.01, 0.1, 0.2), and kernel coefficients (gamma: 'scale', 'auto', 0.001, 0.01, 0.1). The grid search identifies the best hyperparameters as C = 25, epsilon = 0.2, kernel = 'rbf', and gamma = 0.01. However, the performance difference on the testing set between gamma = 0.01 and gamma = 'scale' is minimal, and the latter yields better training performance. Therefore, we select gamma = 'scale' for the final model.

### Deep neural network setup

The feedforward deep neural network (DNN), implemented using the Keras API in TensorFlow (Abadi et al., 2015), consists of interconnected layers through which information flows in one direction, from input to output. It learns complex nonlinear patterns by adjusting internal weights through backpropagation. We test a range of architectures with one or two hidden layers and different neuron counts (16, 64, 128, and 256) to identify a suitable configuration for $dSL$ prediction and to benchmark its performance against SVR. Additionally, we explore different dropout rates (0.2, 0.3, and 0.4), learning rates (0.001, 0.005, 0.01, and 0.05), and batch sizes (16, 32,

and 64) to optimise training stability and generalisation. Based on testing performance across these trials, the final architecture includes an input layer, one hidden dense layer with 64 neurons using a Rectified Linear Unit activation, and a linear output layer for continuous regression. A dropout layer with a rate of 0.1 follows the hidden layer to mitigate overfitting. The model is trained using stochastic gradient descent with a learning rate of 0.01 and momentum of 0.8. Mean squared error (MSE) is used as the loss function. Training is run for 1,000 epochs with a batch size of 64 and early stopping (patience = 32) to prevent overfitting.

### Skill metrics

To evaluate the ML model performances, we employ the mean absolute error (MAE), MSE, the coefficient of determination ($R^2$), and Pearson's correlation coefficient ($r$), as defined in Equations 2–5, respectively.

$$\text{MAE} = \frac{1}{n}\sum_{i=1}^{n}|y_i - \widehat{y}_i| \tag{2}$$

where MAE measures the average magnitude of absolute prediction errors, $y_i$ is the actual values, $\widehat{y}_i$ is the predicted values, and $n$ is the number of samples.

$$\text{MSE} = \frac{1}{n}\sum_{i=1}^{n}(y_i - \widehat{y}_i)^2 \tag{3}$$

where MSE represents the average squared prediction error, penalising larger errors more severely.

$$R^2 = 1 - \frac{\sum_{i=1}^{n}(y_i - \widehat{y}_i)^2}{\sum_{i=1}^{n}(y_i - \overline{y})^2} \tag{4}$$

where $R^2$ quantifies the proportion of variance in the dependent variable explained by the model, and $\overline{y}$ is the mean of actual values.

$$r = \frac{\sum_{i=1}^{n}(y_i - \overline{y})(\widehat{y}_i - \overline{\widehat{y}})}{\sqrt{\sum_{i=1}^{n}(y_i - \overline{y})^2}\sqrt{\sum_{i=1}^{n}(\widehat{y}_i - \overline{\widehat{y}})^2}} \tag{5}$$

where $r$ measures the linear correlation between actual and predicted values, and $\overline{\widehat{y}}$ is the mean of predicted values.

### Results and discussion

### Seasonal changes in the beach morphology

We define the seasons based on meteorological conventions: winter (December–February, DJF), spring (March–May, MAM), summer (June–August, JJA), and autumn (September–November, SON). Table 1 presents the seasonal mean values of individual and clustered storm characteristics and their associated beach changes ($dSL$ and $dV$). We then select the erosional events for further analysis, and Figure 3 shows radar plots of the mean values of storm characteristics ($SP$ and storm count) and associated erosional beach changes ($dV$ and $dSL$) for individual and clustered events across the four seasons, with radial axes scaled to enable finer comparison among the four features.

Winter (DJF) emerges as the most morphodynamically active season, recording the highest number of erosional events (96, including 22 clusterings). It records seasonal mean values for

**Table 1.** Seasonal distribution of individual and clustered storm events (shown in parentheses), including their storm power (Equation 1) and associated impacts on beach morphology: mean shoreline change, $dSL$ [m], and mean volumetric beach change, $dV$ [m³/m]

| | Number of events | | Storm power [m²·h] | | Mean shoreline change [m] | | Mean beach change [m³/m] | |
|---|---|---|---|---|---|---|---|---|
| Season | Erosional | Accretional | Erosional | Accretional | Erosional | Accretional | Erosional | Accretional |
| Winter [DJF] | 74 (22) | 54 (10) | 994.63 (2,584.59) | 852.34 (1,622.35) | 3.23 (8.30) | −1.73 (−2.82) | 13.78 (22.55) | −12.73 (−12.56) |
| Spring [MAM] | 52 (19) | 44 (12) | 1,185.64 (2059.07) | 603.51 (1,319.50) | 3.69 (5.54) | −2.00 (−2.51) | 10.68 (13.36) | −11.06 (−12.59) |
| Summer [JJA] | 40 (7) | 18 (0) | 938.87 (1,615.60) | 1,058.56 (–) | 6.70 (8.20) | −0.11 (–) | 12.99 (21.20) | −5.99 (–) |
| Autumn [SON] | 70 (19) | 53 (7) | 1,336.47 (3,023.30) | 1,285.79 (1,312.19) | 6.10 (11.67) | −1.88 (−1.09) | 16.62 (33.55) | −14.22 (−12.43) |

*Note:* Seasons are defined based on meteorological conventions: winter (December–February), spring (March–May), summer (June–August), and autumn (September–November). Erosional values are considered positive, while negative values denote accretionary events.

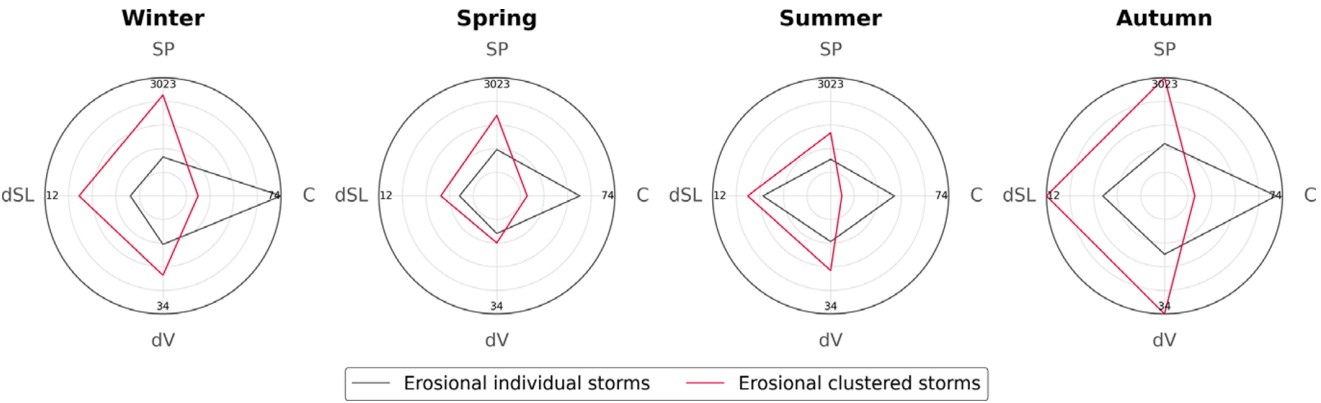

**Figure 3.** Radar plots of normalised mean values of individual and clustered storm characteristics and their associated erosional (excluding accretionary) beach changes across the four meteorological seasons: winter (DJF), spring (MAM), summer (JJA), and autumn (SON). Each axis represents one of the following features, arranged clockwise from the top: storm count ($C$), mean storm power ($SP$; Equation 1), mean shoreline change ($dSL$ [m]), and mean volumetric beach change ($dV$ [m³/m]). Uniform radial scaling across all subplots enables direct seasonal comparison. The highest recorded value of each feature across all seasons is highlighted to aid comparison with the corresponding value in each season.

$SP$ (2,584.59 m²·h), $dSL$ (8.30 m), and $dV$ (22.55 m³/m), all associated with clustered storm events. These patterns align with previous findings (Dorman et al., 2004; Eichentopf et al., 2020), emphasising the role of extratropical cyclones in driving high-energy waves, sustained sediment transport, and shoreline retreat during winter months. Spring (MAM), with transitional characteristics between the intense winter season and the relatively calm summer (JJA), displays moderate activity of erosional events (71, including 19 clusterings), where the seasonal mean value of $SP$ (1,185.64 m²·h) for individual erosional events is significantly higher than that for accretional events (603.51 m²·h). Mean accretionary volumetric beach changes for both individual and clustered events show similar values in the winter and spring seasons.

Autumn (SON) storms, although often underemphasised, play a significant role in shaping coastal geomorphology at Hasaki. While recording slightly fewer erosional events (89, including 19 clusterings) than winter, autumn exceeds all other seasons in storm intensity and beach erosion, with the highest seasonal mean values of $SP$ (3,023.30 m²·h), $dSL$ (11.67 m), and $dV$ (33.55 m³/m), all associated with clustered storm events. The significant beach changes during autumn are largely due to high typhoon activity in September and October, when elevated sea surface temperatures and a weakened Pacific high-pressure system make Japan particularly vulnerable to intense tropical cyclones (Nayak and Takemi, 2023). These findings highlight the importance of considering not just storm frequency but also storm intensity, especially the often

overlooked autumn storm dynamics, when developing coastal management strategies at sites like Hasaki.

As expected, a smaller number of storms, both clustered and individual, occur during the summer. However, the mean $dV$ value for individual events is comparable to that of winter. Due to the limited number of clustered events (7), we do not focus on them when drawing conclusions.

### Role of storm clusters

Figure 4a shows the distribution of erosion magnitudes: $dSL$ and $dV$ (from left to right, respectively). Figure 4a-i and a-ii shows results for individual storms, which generally exhibit lower erosion magnitudes with near-symmetric distributions centred around minimal beach change. Figure 4a-iii and a-iv presents the same analysis for clustered storms, revealing a rightward skew–indicative of asymmetric distributions toward higher erosion magnitudes and underscoring the severe geomorphic impacts of storm clustering.

During 2005–2007, the mean $dSL$ from clustered storms was 11.62 m of retreat, significant but not the highest among all periods, while the associated $dV$ reached 37.71 m³/m of erosion, marking the largest 3-year mean beach change recorded. This highlights the disproportionate impact of storm clustering on different beach zones: while the swash zone may exhibit limited retreat (reflected in moderate $dSL$), overall sediment removal across the 100-m-wide beach remains substantial.

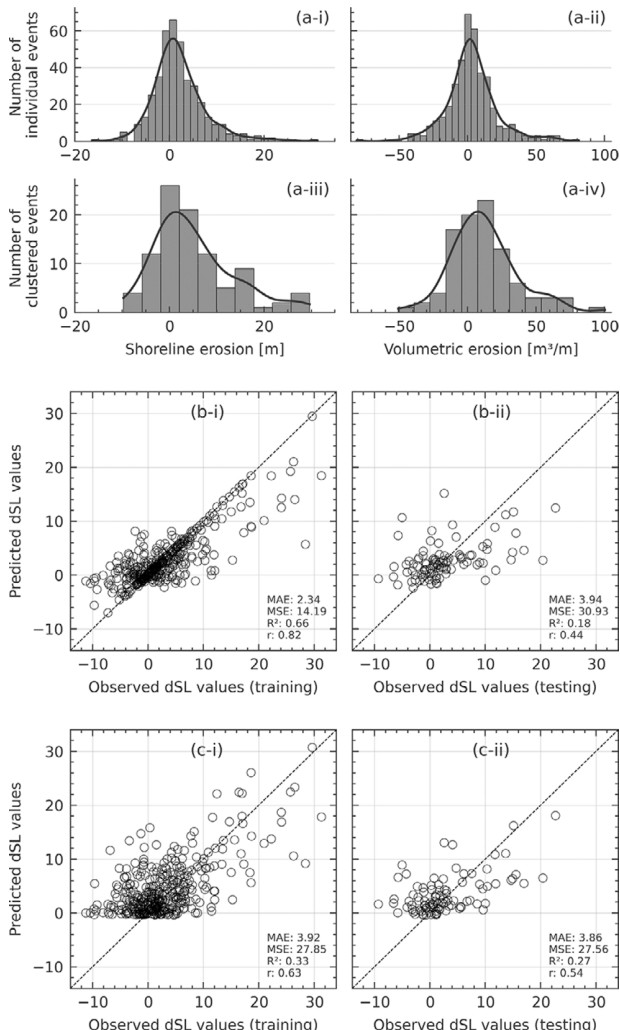

**Figure 4.** Distributions of shoreline change ($dSL$) and volumetric erosion ($dV$) values for individual storms (a-i, ii) and clustered storms (a-iii, iv). Performance of the SVR (b-i and b-ii) and DNN (c-i and c-ii) models on training and testing datasets for predicting shoreline change ($dSL$) across all storm events. Evaluation metrics – MAE, MSE, coefficient of determination ($R^2$), and Pearson's correlation coefficient ($r$) – are provided within (b) and (c).

Similarly, during 2002–2004, clustered storms produced a mean shoreline retreat of 12.13 m ($dSL$) and a mean volumetric erosion of 29.19 m³/m ($dV$), despite the occurrence of only three erosional clustered events. These results demonstrate that $dV$ is a more robust indicator of erosion severity than $dSL$ alone, as it captures the cumulative and directional impacts of closely spaced high-energy events occurring before natural recovery. Supporting this, Harley (2017) emphasised that the internal sequence and directionality of waves within storm clusters can intensify beach erosion beyond the combined effects of individual storms. Supplementary Table A1 provides a comprehensive summary of $SP$ values and the associated beach erosional and accretional impacts of individual and clustered storms.

### Benchmarking ML models

#### Support vector regression
Figure 4b-i and b-ii shows the performance of the SVR model on the training and testing datasets, respectively. The model achieves

reasonable performance on the training dataset (MAE: 2.34 m, MSE: 14.19 m², $R^2$: 0.66, $r$: 0.82). However, its testing performance is substandard (MAE: 3.94 m, MSE: 30.93 m², $R^2$: 0.18, $r$: 0.44), indicating that significant improvements are needed to develop a more generalised predictive application. The clear difference between training and testing performance suggests that the model may be overfitting the training data, limiting its ability to generalise to unseen cases. When using a fixed `sigma` value of 0.01 instead of the `'scale'` parameter for the kernel width, signs of overfitting in the training set disappear, but testing performance remains nearly unchanged. The `'scale'` setting adapts the kernel width based on the data distribution, which can lead to overly complex decision boundaries that fit noise in the training data, thereby causing overfitting.

#### Deep neural network
Unlike SVR, training a DNN model requires an additional validation set during the backpropagation process. Accordingly, 20% of the training data is used for validation (16% of the total wave storms: 64 individual events and 16 clustered events), following standard practice in deep learning workflows. Figure 4c-i and c-ii shows the DNN model's performance on the training and testing datasets, respectively. The model does not achieve satisfactory results on either set. On the training dataset, the DNN yields MAE = 3.92 m, MSE = 27.85 m², $R^2$ = 0.33, and $r$ = 0.63. On the testing dataset, the performance is similarly limited, with MAE = 3.86 m, MSE = 27.56 m², $R^2$ = 0.27, and $r$ = 0.54.

#### Learnt lessons and future directions
Despite trialling multiple train–test splits by varying the `random_state` parameter, both ML models consistently failed to generalise to unseen data. This suggests that the poor performance stems not from sampling variability. Instead, it is due to fundamental limitations in the dataset, particularly the small sample size and a sparse set of predictive features. We optimised SVR hyperparameters via grid search and explored various DNN architectures; therefore, model configuration is unlikely to be the primary issue. The only geomorphic input used in the present analysis was pre-storm shoreline position, which constrained the models' ability to capture key physical processes of nearshore morphodynamics. Our unsuccessful attempts to model $dSL$ further highlight the need for richer, more diverse input data.

As shown by Seenath (2025), simplified physics-based models, such as the one-line theory or the Bruun rule, also struggle to reproduce shoreline behaviour when limited input data are available. To enhance predictive capability, ML approaches should integrate additional geomorphic and hydrodynamic indicators, such as nearshore bathymetry, sediment characteristics, and tidal and current data, which can be obtained through field surveys, remote sensing, or both. For example, recent advancements in remote sensing may enable the extraction of beach cusp or sandbar features, such as wave breaking zones, which reflect key geomorphic processes. Complementary insights from experimental and field-based studies on surf zone sandbar erosion and sea-level-driven morphological change further underscore the importance of accounting for sediment transport dynamics and their interaction with hydrodynamic forcing in predictive modelling (Islam et al., 2024; Enayatighadikolaei et al., 2025). Access to such high-quality, standardised datasets would improve model training and support wider applicability across diverse coastal settings.

Although the current ML models in this analysis underperformed, we consider these results a useful baseline. ML approaches,

particularly when integrated with simplified physical models, can offer efficient, scalable alternatives where resource-intensive models, such as Delft3D or XBeach, are not feasible. ML can extract insights from diverse data sources and help identify patterns not easily captured by traditional methods. However, realising this potential requires improved feature engineering and access to large, openly shared coastal datasets with high spatial and temporal resolution. Therefore, we highlight the importance of open data sharing to advance ML-driven approaches for coastal modelling and beach change prediction.

### Temporal evolution of storm dynamics

Approximately 80% of the identified storm events are classified as individual events, with the majority (60%) leading to erosional impacts (Figure 4a), as would be expected given the energetic nature of such events. From 1987 to 2022, the annual number of storms shows a trend of increase, primarily due to an increase in the frequency of occurrences of severe wave conditions at Hasaki, indicating a possible impact of climate change (Figure 2b) and Supplementary Table A1. Mori et al. (2010), using an atmospheric general circulation model and a global wave model, have highlighted the further intensification of wave heights of the future climate at middle latitudes impacting the Japanese coasts. However, the rise in erosional storm frequency is particularly attributed to an increased occurrence of individual storms rather than clustered events.

$SP$ (Equation 1) quantifications indicate that clustered storms generally exhibit greater power than individual storms, which aligns with their typically longer durations. Over the 36 years, the mean annual $SP$ shows an increasing trend for both individual and clustered storms with erosional impacts on the beach, reflecting a broader intensification of the extreme wave climate. However, exceptionally high values for clustered events are observed in 1996 (5,357.19 m²·h), 1998 (4,257.64 m²·h), and 2014 (7,794.47 m²·h), while such extreme annual means are not evident for individual storm events. Although the $dV$ values for both erosional individual and clustered storms increase with storm power, their $dSL$ values decrease, emphasising that $dV$ is a more reliable indicator of erosion severity than $dSL$, as it captures the cumulative effects of wave sequence and directionality within storm clusters.

### Conclusion

This study reveals the seasonal and temporal complexities of wave storm-induced beach erosion at Hasaki beach, emphasising the disproportionate influence of clustered storm events on shoreline erosion, particularly during high-energy autumn and winter months. While less frequent clustered storm events are not the dominant cause of all erosional events, its cumulative impacts (high-energy wave interactions) are significant and require targeted attention in prediction frameworks. The observed long-term increase in $SP$, alongside stable shoreline change but rising volumetric erosion, suggests that storm energy alone does not fully explain morphological trends.

The poor generalisability of SVR and DNN models underscores the need for more comprehensive predictor features and larger, high-quality datasets, highlighting the importance of open data sharing to advance ML applications in coastal modelling. Despite substandard ML performance, this study lays the groundwork for developing an improved impact-based, seasonal shoreline prediction system, which will be the focus of future research. Integrating detailed geomorphic data, storm dynamics, and seasonal variability will be essential to build robust, generalisable tools that better support coastal management and planning. Overall, these findings underscore the need for seasonally adaptive, morphology-informed forecasting tools to support more effective and resilient coastal management.

**Open peer review.** To view the open peer review materials for this article, please visit http://doi.org/10.1017/cft.2025.10012.

**Supplementary material.** The supplementary material for this article can be found at http://doi.org/10.1017/cft.2025.10012.

**Data availability statement.** The NOWPHAS data (available online at https://www.mlit.go.jp/kowan/nowphas/index_eng.html) were provided by the Ministry of Land, Infrastructure, Transport and Tourism in Japan. The long-term observational dataset of cross-sectional beach profiles taken along the HORS pier (available online at https://www.mlit.go.jp/kowan/nowphas/index_eng.html) was provided by the Port and Airport Research Institute, National Institute of Maritime, Port and Aviation Technology (Banno et al., 2020). The Python scripts used for data pre-processing, storm identification, support vector regression model training, and result visualisations are available upon reasonable request from the corresponding author. They are also intended to be made accessible at https://github.com/salikathilakarathne/storm-clusters.

**Author contribution.** Methodology: S.T.; H.E.; M.S.I.; T.S.; M.M.; R.W. Data curation: S.T. Data visualisation: S.T. Writing original draft: S.T.; H.E.; M.S.I. Writing review & editing: T.S.; M.M.; R.W. All authors approved the final submitted draft.

**Competing interests.** The authors declare none.

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
