## [Reviewer Report]

This manuscript analyses beach erosion along the Hasaki Coast, Japan, using waves as the main source of erosion. It examines how individual storms and storm clusters affect beach profiles differently, finding that clusters tend to cause greater erosion, possibly due to their duration or cumulative energy. The authors also apply machine learning, specifically support vector regression (SVR), using 16 parameters to predict erosion. While the model performs well on the training set, testing results show some limitations. The study is well-structured, clearly written, and presents relevant and valuable findings. I have only a few minor comments that I believe could help strengthen the manuscript before publication.

<b>General comments:</b>

1.While the data and methods clarify that the study focuses on wave-induced erosion, this is not evident in the abstract and introduction, where the term ‘storm beach erosion’ is used. Storms can involve various components—waves, storm surges, wind, and rainfall—all of which may contribute to erosion. I recommend clearly stating that the study isolates the role of waves, since it is the most important factor (with appropriate citation) and ensuring consistency in terminology throughout the abstract and introduction.

2.The location of the Supplementary Material is unclear. While some referenced figures appear in the provided figure folder, a separate Supplementary Material document was not included. As a result, I couldn’t access Table A1.

3.Regarding the SVR section: (1) In line 532, does the random split ensure that the distribution of key characteristics, such as the ratio of individual to clustered storms or storm magnitudes, is preserved across subsets? (2) It may be worth testing an 80–20 split, as 10% might be insufficient for reliable correlation analysis. Additionally, have you considered rotating which storms are assigned to each subset, particularly since some storms appear consistently difficult to model? Repeating the process with different splits could help assess robustness. (3) Finally, have you explored using additional models for comparison?"

<b>Minor comments: </b>

-Wave observations start in 1987; however, wave directional data is only available from 1991 onward. Figure 1 starts in 1987. How do you take care of the 3-year gap in the analysis? If it is by starting the analysis in 1991, I suggest mentioning it in the data section.

-Line 147: “pre-processed to address outliers and missing data”. How? Elaborate briefly

-Lines 152 and 155: Different methods are mentioned. I recommend always explaining, without too many, details those methodologies in the text (in 1 or 2 lines). Same in lines 35-376.

-Line 164: Why 48 hours as an independence window?

-How is the parameter D (duration) defined in clusters? And in individual storms? Time above 1.5 m?

-Eq. 1 and lines 176-177 would fit better in the text after line 185. Right now, those lines feel a bit disconnected from the previous paragraph.

-Line 191: “recorded daily until March 2011…” since?

-Line 266: Normalized by what?

-Lines 312-322: I suggest rewriting this paragraph. For example: “ Figure 4a shows the distribution of erosion magnitudes: dSL and dV (from left to right respectively). Upper panels show the results for individual storms where generally lower magnitudes of erosion… . Lower panels show the same analysis, for clusters, where….”

-Section “Temporal evolution of storm dynamics”, in the provided material there is a figure showing this information, why is it not mention on the text?

---

## [Reviewer Report]

Dear Editorial Board and Authors,

After carefully reviewing the manuscript entitled “Machine learning-aided analysis of storm dynamics and clustering, and their impacts on beach erosion”, I would like to provide the following comments and suggestions:

First, I highly appreciate the authors' significant efforts in collecting and analysing data related to shoreline change at the Hasaki coast observation station in Japan. The successful compilation of a comprehensive dataset with 501 samples is truly commendable. Although this number of samples is not considered large by machine learning standards, it is sufficient for certain machine learning models to potentially achieve good predictive performance.

In this study, the authors applied a machine learning model to predict distributions of shoreline change (dSL), namely the Support Vector Regression (SVR) model. SVR is a commonly used regression model, well-suited for small and relatively simple datasets. However, in this case, the model’s performance on both the training set (R² = 0.72) and testing set (R² = 0.20) remains considerably low. Although the authors have provided some justifications, such results are still unsatisfactory from a scientific and application perspective.

For this study, I would suggest that the authors experiment with alternative machine learning models to improve predictive performance, such as Deep Neural Networks (DNN) or Random Forests. DNNs, in particular, have demonstrated strong capabilities in capturing complex, nonlinear relationships within structured datasets. For small datasets like this, the authors can consider using fewer hidden layers and a limited number of neurons to avoid overfitting. Additionally, incorporating key conditional variables to reduce data dimensionality may also help improve the model’s generalisation ability.

---

## [Editor Report]

Dear Authors,

Thank you for submitting your manuscript “Machine learning-aided analysis of storm dynamics and clustering, and their impacts on beach erosion” to our special issue.

After careful consideration of the two independent peer reviews, I would like to share my editorial decision and comments.

The two reviewers provided differing recommendations: one suggested minor revision, while the other recommended rejection. Although this may seem contradictory at first glance, I find myself agreeing with the key points raised by both reviewers. Each of them has provided valuable insights that, when taken together, highlight both the strengths and the critical weaknesses of the manuscript.

Based on these considerations, my recommendation is major revision.

In agreement with Reviewer #2, I share the concern that the machine learning model, as currently presented, does not achieve satisfactory predictive performance to robustly support the paper’s central claims. Although the authors acknowledge this limitation, they proceed with their analyses and draw conclusions without sufficiently contextualizing these results in light of the model’s performance. As a result, the validity of the conclusions is potentially compromised and should be reconsidered or more carefully framed.

I do not wish to overstep, as ultimately the approach and framing are decisions for the authors to make. However, I would like to offer several options the authors may want to consider to strengthen the manuscript:

1. Further test and refine the current model to improve its performance, possibly by addressing overfitting or by incorporating additional features in the model.

2. Explore alternative machine learning models to assess whether other approaches can achieve better generalization, in agreement with Reviewer #2. This could be pursued alongside option 1.

3. Reframe the paper as a negative result, explicitly stating that the current machine learning approach was unable to produce a satisfactory predictive model for shoreline change, which in itself is a valuable contribution to the field.

Additionally, I encourage the authors to treat model performance and validation as a core component of the manuscript. This would benefit from a dedicated, clearly structured section in both the Methods and the Results. At present, these aspects are somewhat mixed and would benefit from more focused and transparent reporting. These revisions should be addressed alongside the comments provided by both reviewers, who carefully evaluated the manuscript.

I believe this manuscript can make a valuable contribution once these substantial revisions are addressed.

Thank you for your submission. I look forward to receiving a revised version of your manuscript.

Best regards,

Alex R Enriquez

---

## [Reviewer Report]

-MAIN QUESTION: Since the authors have changed the models 80-20 split and added a new one. I wonder if the figures, results and the numerical values presented in the paper have been updated accordingly.

-Line 20 (track changes): The word “develop” if used twice in the same sentence, consider replacing.

-Machine learning models and trains section: I suggest revising the first sentence to read: We use two machine learning models, SVR… This would help set the context for the subsections that follow.

-Lines 542-572 (track changes): This paragraph would be more appropriate for the discussion/conclusion section.

---

## [Reviewer Report]

Dear Editors and Authors,

First, I would like to sincerely acknowledge your efforts in applying machine learning to the assessment and prediction of coastal erosion, particularly under the constraints of limited data availability. This challenge is not only specific to coastal erosion studies but is also common across natural hazard research more broadly. I also appreciate the authors’ efforts in revising the manuscript thoroughly in response to the reviewers’ comments.

It is evident that by incorporating the DNN model as a benchmark, the evaluation results on the test set have improved. Although the predictive performance remains relatively modest, I believe that this revised manuscript makes a meaningful scientific contribution. Importantly, it provides a valuable foundation for future research aiming to further enhance the predictive performance of coastal erosion modeling.

In light of the substantial improvements and clarifications made in this revision, I recommend that the manuscript now be considered for publication.

Sincerely,

---

## [Editor Report]

The authors have addressed the concerns of the reviewers and editors. The paper is accepted for publication.